# Factors Affecting Delirium in ICU Patients

**DOI:** 10.3390/ijerph20105889

**Published:** 2023-05-19

**Authors:** I Seul Jeong, Mi-Kyoung Cho

**Affiliations:** Department of Nursing Science, College of Medicine, Chungbuk National University, Cheongju 28644, Republic of Korea; dew900605@cbnuh.or.kr

**Keywords:** delirium, intensive care unit, blood urea nitrogen, length of stay, Glasgow coma scale, intensive care unit, restraints, simplified acute physiology score, morse fall scale

## Abstract

This study examined delirium severity using a delirium screening tool and analyzed the predictors, including pain, acuity, level of consciousness, fall risk, and pain score, to increase understanding of delirium and present foundational data for developing nursing interventions for delirium prevention. This was a retrospective study of 165 patients admitted to three intensive care units (ICUs). the Nursing Delirium Screening Scale (Nu-DESC) was used as a research tool to screen for delirium and measure the degree of delirium. The incidence of delirium in patients was 53.3%, and the average delirium score in the delirium group was 2.40 ± 0.56. Nu-DESC scores were significantly correlated with ICU days, ventilator days, restraint applications, the number of catheters inserted, sedative medication use, the Simplified Acute Physiology Score (SAPS III), the Morse Fall Scale (MFS), the Glasgow Coma Scale (GCS) scores, pain scores, and blood urea nitrogen (BUN). Stepwise multiple linear regression showed that the number of restraint applications, GCS score, ICU days, and BUN levels were factors influencing delirium. Based on the findings, ICU nurses should use delirium screening tools to ensure accurate delirium screening and work to reduce the incidence and degree of delirium by observing factors affecting delirium in patients.

## 1. Introduction

### Rationale for the Study

Delirium is a multifactorial neuropsychiatric syndrome with an abrupt onset, characterized by symptoms such as an altered level of consciousness, loss of memory and orientation, diminished environmental perception, and cognitive decline [1]. Other symptoms include loss of concentration, nonsystematic thinking, changes in the level of consciousness and acute mental status, cognitive impairment, disruption of the sleep-wake cycle, and language disorder; the symptoms may undergo several cycles of improvement and exacerbation even in one day [2]. 

The incidence of delirium in the intensive care unit (ICU) ranges from 7.4–81.7% [3,4,5], and delirium most commonly occurs on days 1–2 after ICU admission [6]. Furthermore, delirium impacts patients’ prognosis by extending the duration of mechanical ventilation and increasing the mortality rate among ICU patients [7]. 

Delirium can be quickly resolved once the cause is eliminated, thus highlighting the importance of prevention and early detection over treatment [8]. In Korea, the Confused Assessment Method for the ICU (CAM-ICU) and the Nursing Delirium Screening Scale (Nu-DESC) are the most widely used instruments for assessing delirium [9]; however, only 9–12% use delirium screening tools, and the reported interrater disagreement rate is 18.7% [10,11].

Delirium is influenced by patient history pre-hospitalization factors such as age, education level, visual acuity, and hearing impairment [12], and the use of psychiatric medications before hospitalization [13]. Factors that trigger delirium after hospital admission include the acute physiology and chronic health evaluation II (APACHE II) score, the Sequential Organ Failure Assessment (SOFA) score, physical restraints, use of a ventilator, hypoalbuminemia, acidosis, length of stay in the ICU, use of anticonvulsants, orthopedic surgery, postoperative use of antipsychotics, a nasogastric tube, a urinary catheter, and the insertion and retaining of a central venous catheter [4,6,10,14,15]. In addition, falls are significantly correlated with delirium [16] Pain scores in delirium patients are higher than those in no-delirium patients [17,18]. Furthermore, immobility and the use of restraints, witnessing another patient’s death, an unfamiliar environment, and a 24-h lighted environment in the ICU also contribute to the onset of delirium [19,20]. Regarding hematological parameters, delirium patients have lower protein, albumin, hemoglobin, and potassium [21] levels, but higher lactic acid, C-reactive protein (CRP), and creatinine levels [22].

Although there has been continued research on the factors contributing to the onset of delirium in Korea and abroad, the findings have been inconsistent due to variations in participant characteristics, environmental factors, and types of ICU across the studies. As the Nu-DESC is a better instrument than the CAM-ICU for assessing the severity of delirium based on delirium screening and score, the present study aimed to assess the severity of delirium among adult patients admitted to the ICU using Nu-DESC and identify the predictors of delirium.

## 2. Materials and Methods

### 2.1. Study Design

This study is a retrospective survey aiming to identify the predictors of the severity and onset of delirium in ICU patients. 

### 2.2. Study Population

Patients admitted to one of three ICUs (Emergency Department ICU [EICU], general ICU [ICU], and trauma ICU [TICU]) at a tertiary hospital in Cheongju, South Korea, between 15 May and 31 August 2022, were enrolled. The sample size was calculated using the G*power program 3.1.9.4 (Heinrich-Heine-Universität Düsseldorf, Düsseldorf, Germany) to achieve a medium effect size of 0.15 based on Cohen [23], an alpha of 0.05, and a power of 0.8 with 19 variables and a multiple linear regression: fixed model and *R*^2^ deviation from zero. The minimum sample size was calculated to be 153. With a 7% anticipated withdrawal rate, we reviewed the medical records of 165 patients.

### 2.3. Instruments

#### 2.3.1. Delirium

Delirium was assessed using the Nu-DESC developed by Gaudreau et al. [24] and adapted into Korean by Kim et al. [25]. The instrument consists of five items (disorientation, inappropriate behavior, inappropriate communication, illusions/hallucinations, and psychomotor retardation), and each item is rated as 0 (absent) or 1 (present), with the total score ranging from 0–5. The cutoff point for delirium is 2, where a score of 0–1 indicates no delirium, while a score of 2 or higher indicates delirium. At the time of development by Gaudreau et al. [24], the scale had a sensitivity of 0.85 and a specificity of 0.86. In the study by Kim et al. [25], the Korean version of the Nu-DESC had a sensitivity of 0.81 and a specificity of 0.97. In the present study, the scale had a sensitivity of 0.82 and a specificity of 0.93.

#### 2.3.2. Predictors of Delirium

To identify the predictors of delirium, we investigated the factors previously reported to be associated with delirium in terms of general, clinical, and hemodynamic characteristics. Four general characteristics were investigated: sex, age, pre-hospitalization drinking status, and education level. Eleven clinical characteristics were investigated: ICU length of stay (LOS), use and duration of mechanical ventilation, use and number of physical restraints, number of catheters, number of sedatives, acuity, fall risk score, level of consciousness, and pain score. In addition, four hemodynamic characteristics were investigated: blood urea nitrogen (BUN), high sensitivity CRP (hs-CRP), white blood cell (WBC), and lactic acid. Among clinical characteristics, acuity was assessed based on the Simplified Acute Physiology Score (SAPS III) [26] and based on a cutoff of 41 [27]; a score of 41 or higher indicates high acuity. In the present study, fall risk was assessed using the Morse Fall Scale (MFS) [28], and based on a cutoff of 45; a score of 45 or higher indicates high fall risk. Level of consciousness was assessed using the Glasgow Coma Scale (GCS) [29] and based on a cutoff of 13 [30]; a score of below 13 indicates a state of not being fully conscious. In this study, the pain was assessed based on a numeric rating scale [31], and a face pain rating scale (FPRS) [32]. Both scales use a scoring system from 0–10. According to the WHO Pain Management Guideline [33], a score of 4 or higher indicates “moderate or severe” pain. 

### 2.4. Data Collection

One hundred and sixty-five patients admitted to the EICU, ICU, and TICU at a tertiary hospital in Cheongju, South Korea, between 15 May and 31 August 2022, with at least 24 h elapsed from the time of admission, were analyzed. Before data collection, the study was approved by the Institutional Review Board (IRB) (IRB No. 2022-05-020-001) as ‘subjects were exempted from informed consent’. Delirium was assessed by the author using the Korean version of the Nu-DESC, around 3–4 PM in 2-day intervals (post-admission days 2, 4, 6) with reference to the study by Klouwenberg et al. [34]. General characteristics, clinical characteristics, and hemodynamic characteristics were obtained from electric medical records (EMR). Among clinical characteristics, the fall risk score, acuity, the pain score, GCS, and hemodynamic parameters were taken from the day of the delirium assessment, and average values were used. 

### 2.5. Data Analysis

The collected data were analyzed using SPSS 26.0 for Windows (IBM Corp., Armonk, NY, USA). For the general, clinical, and hemodynamic characteristics, incidence, and severity of delirium, categorical variables are presented as real numbers and percentages, and continuous variables are presented as means with standard deviations and minimum and maximum values. Normality was tested using the Kolmogorov–Smirnov test.

The differences between the delirium and no-delirium groups and the comparison of delirium severity were analyzed using independent *t*-test, χ^2^ test, and one-way ANOVA. In addition, the relationships between delirium severity and continuous variables were analyzed using Pearson’s correlation. Finally, the predictors of delirium were analyzed with stepwise linear multiple regression analysis. 

## 3. Results

### 3.1. Participants’ Characteristics

Of 165 participants, 96 (58.2%) were male, and the mean age was 68.7 (standard deviation [SD]: 15.42) years. The mean ICU LOS was 11.66 (SD: 9.42) days, and the mean duration of mechanical ventilation was 5.72 (SD: 9.88) days; sixty-five (39.4%) were currently on a ventilator, and 96 (58.2%) were on physical restraints. The average number of physical restraints applied was 1.53 (SD: 1.53), and the mean number of catheters placed was 3.84 (SD: 1.37). The most common catheter was a Foley catheter (93.9%), followed by an arterial line (92.1%), central line (67.9%), and nasogastric tube (61.8%). The mean number of sedatives used was 0.45 (SD: 0.63), and the mean SAPS III score was 52.66 (SD: 14.14). The mean MFS score was 50.99 (SD: 9.45), and the mean GCS score was 12.55 (SD: 1.97). The mean pain score was 0.97 (SD: 1.51). The mean BUN was 29.88 (SD: 17.90) mg/dL, and the mean lactic acid concentration was 1.83 (SD: 1.78) mmol/L (Table 1).

### 3.2. Severity and Incidence of Delirium

Of 165 participants, 88 (53.3%) had delirium, and the mean delirium score in the delirium group was 2.40 (SD: 0.56) (Table 2).

### 3.3. Comparison of Characteristics between Delirium and No-Delirium Groups

The mean ICU LOS (*t* = −2.63, *p* = 0.009) and duration of mechanical ventilation (*t* = −4.46, *p* < 0.001) were significantly longer in the delirium group than the no-delirium group, and the mean number of physical restraints (*t* = −8.00, *p* < 0.001), number of catheters placed (*t* = −5.73, *p* < 0.001), and number of sedatives used (*t* = −4.42, *p* < 0.001) were higher in the delirium group than in the no-delirium group. The percentages of participants with a ventilator (*X*^2^ =15.51, *p* < 0.001) and physical restraints (*X*^2^ = 52.03, *p* < 0.001) were higher in the delirium group than in the no-delirium group. The mean SAPS III score (*t* = −3.72, *p* = 0.002) and BUN (*t* = −3.80, *p* < 0.001) were higher in the delirium group, but the mean GCS score (*t* = 8.79, *p* < 0.001) and pain score (*t* = 1.98, *p* = 0.049) were lower in the delirium group than the no-delirium group. There were no significant differences between the two groups in the general characteristics, MFS score, CRP levels, and WBC count (Table 3).

### 3.4. Differences in Delirium According to Participant Characteristics

The Nu-DESC score was higher in patients with ≥13 days in the ICU (*t* = −2.33, *p* = 0.027), ≥ 6 days on a ventilator (*t* = −4.46, *p* < 0.001), a SAPS Ⅲ score ≥ 41 (*t* = −2.88 *p* = 0.005), an MFS score ≥ 45 (*t* = −3.01, *p* = 0.003), use of a ventilator (*t* = 3.78, *p* < 0.001), use of physical restraints (*t* = 10.16, *p* < 0.001), use of sedatives (*t* = 4.42, *p* < 0.001), use of more than two physical restraints (*t* = −5.47, *p* < 0.001), insertion of more than four catheters (*t* = −5.62, *p* < 0.001), a GCS score < 13 (*t* = 6.65, *p* < 0.001), a pain score < 4 (*t* = 2.69, *p* = 0.008), and lactic acid level < 0.5 mmol/L or > 1.6 mmol/L (*t* = −2.08, *p* = 0.040) (Table 4).

### 3.5. Correlations between Delirium and Participant Characteristics

Delirium was significantly correlated with ICU LOS (*r* = 0.70, *p* < 0.001), duration of mechanical ventilation (*r* = 0.31, *p* < 0.001), number of physical restraints (*r* = 0.63, *p* < 0.001), number of catheters placed (*r* = 0.45, *p* < 0.001), number of sedatives used (*r* = 0.33, *p* < 0.001), SAPS III (*r* = 0.30, *p* < 0.001), MFS (*r* = 0.19, *p* = 0.013), GCS (*r* = −0.54, *p* < 0.001), pain (*r* = −0.22, *p* < 0.001), and BUN (*r* = 0.22, *p* = 0.004) (Table 5).

### 3.6. Predictors of Delirium

To identify the predictors of delirium severity, variables that were significant in the univariate analysis were further analyzed. Nominal variables (use of a ventilator, use of physical restraints) were dummy-coded, and continuous variables (LOS, duration of mechanical ventilation, number of physical restraints, number of catheters, number of sedatives, SAPS III, GCS, pain score, BUN, lactic acid) were entered as is for stepwise multiple regression. The regression model explained 53.0% of the variance and was statistically significant (*F* = 5.70, *p* = 0.018). The Durbin-Watson statistic was close to 2, at 1.95, confirming the independence of residuals. Tolerance was below 1 (0.80–0.98), and the variance inflation factor (VIF) was below 10 (1.02–1.24), confirming the absence of multicollinearity. 

The results showed that the number of physical restraints (*β* = 0.46, *p* < 0.001), GCS score (*β* = −0.34, *p* < 0.001), ICU LOS (*β* = 0.14, *p* = 0.010), and BUN (*β* = 0.13, *p* = 0.018) significantly predicted delirium. That is, the risk of delirium was higher with a greater number of physical restraints applied, decreasing the GCS score, and increasing ICU LOS and BUN (Table 6).

## 4. Discussion

This study examined delirium severity using a delirium screening tool and analyzed the predictors, including pain, acuity, level of consciousness, fall risk, and pain score, to increase understanding of delirium and present foundational data for developing nursing interventions for delirium prevention. 

In this study, delirium severity was assessed using the Nu-DESC, and the incidence of delirium was 53.3%. In a previous study, the incidence of delirium among trauma patients in the TICU was 34.8% [35]. Because trauma patients have altered consciousness (coma and semi-coma) due to head injuries and hemorrhage, there would have been cases in which delirium could not be assessed. Among medical ICU patients, the incidence of delirium has been reported to be 69.0% [36]. The higher incidence may be because a higher percentage of medical ICU patients are on a ventilator or sedated compared to TICU patients, and the risk of delirium would increase with increasing ICU LOS and patient acuity. The present study was conducted on patients in an ICU for both medical and trauma patients, so the incidence of delirium was average and did not significantly differ from previous studies.

In the present study, the delirium group had a higher percentage of patients on a ventilator and a greater duration of mechanical ventilation. Furthermore, delirium severity was higher among patients on a ventilator and among patients who had been on mechanical ventilation for more than 6 days. This finding is consistent with the findings of Ahn et al. [4], where the percentage of patients on a ventilator was higher in the delirium group than the no-delirium group, and that of Kooken et al. [13], where the delirium group was on mechanical ventilation longer on average than the no-delirium group. The mean number of catheters placed in our participants was 3.84, and the most common catheter was a Foley catheter, followed by the arterial line, central line, and nasogastric tube. Similar findings were observed by Park [36], where the delirium group most commonly had an arterial line, followed by a Foley catheter, a gastrointestinal tube, and an artificial airway. The delirium group had more catheters placed on average, consistent with previous studies [6,37]. Thus, the severity of delirium tends to increase with the increasing duration of artificial ventilation and the increasing number of physical restraints and catheters placed. It could be that using a ventilator and having multiple catheters and physical restraints limit a patient’s range of activities, which in turn causes disorientation to time and space, thereby altering their level of consciousness and causing delirium. Thus, removing unnecessary catheters and physical restraints would be important based on the patient’s status. 

In terms of sedatives, the delirium group was given more sedatives than the no-delirium group, and the severity of delirium was higher among those who had been sedated. Kim and Ahn [35] also reported that 87.5% of their delirium group had been sedated, and Seo [37] reported that the use of sedatives such as midazolam and lorazepam are triggers of delirium. Patients placed on a ventilator are mildly sedated to minimize the risk of bumping into the ventilator and to enhance compliance. Thus, this result aligns with the significant difference in the use of a ventilator between the two groups. In light of these findings, quick ventilator and sedative weaning are important, and if long-term use of sedatives is indicated, it would be necessary to replace the drugs with non-opioid sedatives. The SAPS III score, which represents acuity, was higher in the delirium group, consistent with the findings of Sieber et al. [27]. 

Stepwise multiple regression was performed using the significant variables in the univariate analysis. The results showed that ICU LOS, the number of physical restraints, GCS score, and BUN significantly predicted delirium. ICU LOS has been reported in previous studies to be longer in delirium patients than no-delirium patients [13,38] and to be a predictor of delirium [4,13]. An ICU has lights on 24 h a day, has no windows, and treatment and tests are performed around the clock. Thus, there is no distinction between day and night, and patients become disoriented to place and time, which in turn seems to contribute to the onset of delirium. In the present study, 58.2% of patients were on physical restraints, and the percentage was higher in the delirium group than in the no-delirium group. The percentage of patients on physical restraints was lower in previous studies, at 34.3% in the study by Kim and Park [39] and 43% in the study by Perren et al. [40]. 

In the Kim and Park study [39], the most common reason for nurses to apply physical restraints on patients was to “ensure continued placement of a medical device”. Once delirium develops, patients become disoriented to place and attempt to remove anything on their body that causes discomfort. Hence, patient monitors and catheters for drug administration and treatment could be prematurely removed by patients. Thus, the percentage of patients with physical restraints is expected to be higher, with a higher incidence of delirium. However, because the use of physical restraints has been reported to not be effective for the prevention of unplanned extubation [41], physical restraints should only be applied when necessary, as opposed to prophylactically using them to prevent delirium. 

Regarding the relationship between GCS and the incidence of delirium, Bryczkowski et al. [42] reported that GCS influences the onset of delirium, and the Clinical Practice Guidelines by the Society of Critical Care Medicine (SCCM) [43] also pinpointed disorders of consciousness as a major cause of delirium. Maneewong et al. [44] also reported that the risk for delirium increases with decreasing GCS scores. When assessing GCS, a low score for verbal response is given for patients disoriented to time, place, and person, so patients with delirium would probably be given a lower GCS score than those without delirium. Therefore, ICU nurses should pay careful attention to patients’ level of consciousness and orient them as necessary. BUN has been identified as a predictor of delirium, and previous studies have reported that acute renal failure increases the risk of delirium ten-fold [45], and that high BUN and creatinine increase the risk of delirium by 1.67 times [46]. Wan et al. [47] and Pang et al. [48] reported that elevated BUN indicates renal dysfunction and injury, which causes the ineffective elimination of protein metabolites (e.g., uric acid, ammonia) through the kidneys and the consequent accumulation of these wastes in the body. This accumulation results in neurotoxicity and alters a patient’s level of consciousness, such as causing overexcitement and abnormal epileptic activity, thereby contributing to the onset of delirium.

The present study identified the predictors of delirium in patients in the ICU and confirmed previously reported predictors. Based on these results, ICU nurses should be aware of and carefully monitor these potential predictors to provide nursing interventions to prevent the onset of delirium. 

## 5. Conclusions and Recommendations

This study investigated the incidence and predictors of delirium among patients in the ICU using the Korean version of the Nu-DESC. The results showed that the risk for delirium was higher with a greater number of physical restraints, a decreasing GCS score, and increasing ICU LOS and BUN. 

Due to the retrospective nature of this study, some factors associated with delirium may not have been examined because they were not available on medical records, such as environmental factors (e.g., lighting, noise, and sleep). Consequently, prospective studies that include environmental factors are needed.

## Figures and Tables

**Table 1 ijerph-20-05889-t001:** Characteristics of the participants (*n* = 165).

Characteristic		*n* (%)	M ± SD	Range
Sex	Male	96 (58.2)		
	Female	69 (41.8)		
Age (year)			68.70 ± 15.42	19~92
	<65.0	52 (31.5)		
	65.0~74.9	42 (52.5)		
	75.0~84.9	54 (32.7)		
	≥85.0	17 (10.3)		
Drinking	Yes	32 (19.4)		
	No	133 (80.6)		
Education	Less than middle school	89 (53.9)		
	More than high school	76 (46.1)		
Days of ICU stay			11.66 ± 9.42	3~54
Use of ventilation	Yes	65 (39.4)		
	No	100 (60.6)		
Duration of ventilation use (day)			5.72 ± 9.88	0~52
Use of restraint	Yes	96 (58.2)		
	No	69 (41.8)		
Number of restraints			1.53 ± 1.53	0~5
Number of catheters			3.84 ± 1.37	0~6
Type of catheter *	Foley catheter	155 (93.9)		
	Arterial catheter	152 (92.1)		
	Central line	112 (67.9)		
	Nasogastric tube	102 (61.8)		
	Artificial airway	72 (43.6)		
	Hemodialysis catheter	40 (24.2)		
Number of sedative drugs			0.45 ± 0.63	0~2
SAPS Ⅲ score			52.66 ± 14.14	25~85
Morse Fall scale score			20.99 ± 9.45	35~75
GCS score			12.55 ± 1.97	5~15
Pain score			0.97 ± 1.51	0~7
BUN (mg/dL)			29.88 ± 17.90	
CRP (mg/dL)			9.57 ± 7.08	
WBC (10^3^/μL)			11.75 ± 5.53	
Lactic acid (mmol/L)			1.83 ± 1.78	

Notes. M: mean, SD: standard deviation, ICU: intensive care unit, SAPS Ⅲ: simplified acute physiology score Ⅲ, GCS: Glasgow coma scale, BUN: blood urea nitrogen, CRP: C-reactive protein, WBC: white blood cell. * Multiple choice

**Table 2 ijerph-20-05889-t002:** Level of the delirium and occurrence of the delirium (*n* = 165).

Characteristic	Nu-DESC		
	*n* (%)	M ± SD	Range
Non-delirium	77 (46.7)	0.49 ± 0.50	0∼1
Delirium	88 (53.3)	2.40 ± 0.56	2∼4

Notes. Nu-DESC: Nursing Delirium Screening Scale, M: mean, SD: standard deviation.

**Table 3 ijerph-20-05889-t003:** Differences between the delirium group and non-delirium group according to characteristics (*n* = 165).

Characteristic		Non-Delirium (*n* = 77)	Delirium (*n* = 88)	t/χ^2^ (*p*)
		*n* (%) or M ± SD	*n* (%) or M ± SD	
Sex	Male	47 (61.0)	49 (55.7)	0.48 (0.486)
	Female	30 (39.0)	39 (44.3)	
Age (year)		67.32 ± 16.00	69.90 ± 14.88	−1.07 (0.286)
Drinking	Yes	18 (23.4)	14 (15.9)	1.47 (0.226)
	No	59 (76.6)	74 (84.1)	
Education	Less than middle school	43 (55.8)	46 (52.3)	0.21 (0.646)
	More than high school	34 (44.2)	42 (47.7)	
Days of ICU stay		7.58 ± 4.35	12.25 ± 11.10	−2.63 (0.009)
Use of ventilation	Yes	18 (23.4)	47 (53.4)	15.51 (<0.001)
	No	59 (76.6)	41 (46.6)	
Duration of ventilation use (day)		2.38 ± 5.63	8.65 ± 11.73	−4.46 (<0.001)
Use of restraint	Yes	22 (28.6)	74 (84.1)	52.03 (<0.001)
	No	55 (71.4)	14 (15.9)	
Number of restraints		0.66 ± 1.14	2.28 ± 1.42	−8.00 (<0.001)
Number of catheters		3.25 ± 1.23	4.36 ± 1.27	−5.73 (<0.001)
Number of sedative drugs		0.23 ± 0.48	0.64 ± 0.68	−4.42 (<0.001)
SAPS Ⅲ score		48.44 ± 13.25	56.35 ± 13.93	−3.72 (<0.001)
Morse Fall Scale score		49.55 ± 10.34	52.25 ± 8.46	−1.85 (0.067)
GCS score		13.71 ± 1.29	11.52 ± 1.89	8.79 (<0.001)
Pain score		1.22 ± 1.68	0.75 ± 1.32	1.98 (0.049)
BUN (mg/dL)		24.58 ± 13.28	34.53 ± 20.08	−3.80 (<0.001)
CRP (mg/dL)		8.51 ± 6.90	10.49 ± 7.15	−1.81 (0.072)
WBC (10^3^ /μL)		10.98 ± 4.89	12.42 ± 5.98	−1.68 (0.095)
Lactic acid (mmol/L)		1.63 ± 1.72	2.01 ± 1.82	−1.39 (0.166)

Notes. M: mean, SD: standard deviation, t: independent t-test, χ^2^: Chi-square test, ICU: intensive care unit, SAPS Ⅲ: simplified acute physiology score Ⅲ, GCS: Glasgow coma scale, BUN: blood urea nitrogen, CRP: C-reactive protein, WBC: white blood cell.

**Table 4 ijerph-20-05889-t004:** Differences in delirium according to the characteristics (*n* = 165).

Characteristic		*n* (%)	M ± SD	t/F (*p*)
Sex	Male	96 (58.2)	1.43 ± 1.03	−1.14 (0.256)
	Female	69 (41.8)	1.62 ± 1.16	
Age (year)	<65.0	52 (31.5)	1.41 ± 1.16	0.93 (0.428)
	65.0∼74.9	42 (25.5)	1.40 ± 1.01	
	75.0∼84.9	54 (32.7)	1.56 ± 1.08	
	≧85.0	17 (10.3)	1.88 ± 1.11	
Drinking	Yes	32 (19.4)	1.31 ± 1.06	−1.14 (0.258)
	No	133 (80.6)	1.56 ± 1.10	
Education	Less than middle school	89 (53.9)	1.49 ± 1.12	−0.19 (0.852)
	More than high school	76 (46.1)	1.53 ± 1.06	
Days of ICU stay (day)	<12	111 (67.3)	1.38 ± 1.05	−2.33 (0.027)
	≧12	54 (32.7)	1.78 ± 1.13	
Use of ventilation	Yes	65 (39.4)	1.89 ± 0.92	3.78 (<0.001)
	No	100 (60.6)	1.26 ± 1.13	
Duration of ventilation use (day)	<6	116 (70.3)	1.28 ± 1.09	−4.46 (<0.001)
	≧6	49 (29.7)	2.06 ± 0.88	
Use of restraint	Yes	96 (58.2)	2.08 ± 0.90	10.16 (<0.001)
	No	69 (41.8)	0.71 ± 0.79	
Number of restraints	≦2	139 (84.2)	1.32 ± 1.02	−5.47 (<0.001)
	>2	26 (15.8)	2.50 ± 0.91	
Number of catheters	≦4	99 (60.0)	1.15 ± 1.09	−5.62 (<0.001)
	>4	66 (40.0)	2.05 ± 0.85	
Use of a sedative drug	Yes	62 (37.6)	1.97 ± 1.01	4.42 (<0.001)
	No	103 (62.4)	1.23 ± 1.05	
SAPS Ⅲ score	<41	33 (20.0)	1.03 ± 1.19	−2.88 (0.005)
	≧41	132 (80.0)	1.63 ± 1.04	
Morse Fall Scale score	<45	38 (23.0)	1.05 ± 0.99	−3.01 (0.003)
	≧45	127 (77.0)	1.65 ± 1.09	
GCS score	<13	69 (41.8)	2.01 ± 0.89	6.65 (<0.001)
	≧13	96 (58.2)	1.08 ± 1.02	
Pain score	<4	150 (90.9)	1.58 ± 1.08	2.69 (0.008)
	≧4	15 (9.1)	0.80 ± 1.01	
BUN (mg/dL)	6.0∼20.0	57 (34.5)	1.42 ± 1.19	0.75 (0.453)
	<6.0 or >20.0	108 (65.5)	1.56 ± 1.04	
CRP (mg/dL)	≦0.5	3 (1.8)	1.00 ± 1.00	0.82 (0.416)
	>0.5	162 (98.2)	1.52 ± 1.09	
WBC (10^3^/μL)	4.0∼10.0	61 (37.0)	1.43 ± 1.15	0.75 (0.457)
	<4.0 or >10.0	104 (63.0)	1.56 ± 1.06	
Lactic acid (mmol/L)	0.5∼1.6	111 (67.3)	1.39 ± 1.12	2.08 (0.040)
	<0.5 or >1.6	54 (32.7)	1.76 ± 1.00	

Notes. M: mean, SD: standard deviation, t: independent t-test, F: one-way ANOVA, ICU: intensive care unit, SAPS Ⅲ: simplified acute physiology score Ⅲ, GCS: Glasgow coma scale, BUN: blood urea nitrogen, CRP: C-reactive protein, WBC: white blood cell, Normal range: BUN: 6.0∼20.0 mg/dL, CRP: 0∼0.5 mg/dL, WBC: 4.0∼10.0 × 10^3^ /μL, Lactic acid: 0.5∼1.6 mmol/L.

**Table 5 ijerph-20-05889-t005:** Correlation between Nu-DESC scores and characteristics of participants (*n* = 165).

Variable	1	2	3	4	5	6	7	8	9	10	11	12	13
	r (p)												
1	1												
2	0.05 (0.519)	1											
3	0.22 (0.004)	0.05 (0.563)	1										
4	0.31 (<0.001)	0.05 (0.562)	0.13 (0.095)	1									
5	0.63 (<0.001)	0.08 (0.296)	0.12 (0.137)	0.29 (<0.001)	1								
6	0.45 (<0.001)	0.11 (0.177)	0.09 (0.262)	0.49 (<0.001)	0.42 (<0.001)	1							
7	0.33 (<0.001)	0.05 (0.503)	0.10 (0.211)	0.56 (<0.001)	0.41 (<0.001)	0.46 (<0.001)	1						
8	0.30 (<0.001)	0.47 (<0.001)	0.03 (0.692)	0.26 (0.001)	0.31 (<0.001)	0.51 (<0.001)	0.39 (<0.001)	1					
9	0.19 (0.013)	0.21 (0.006)	−0.04 (0.588)	0.25 (0.001)	0.21 (0.008)	0.36 (<0.001)	0.25 (0.001)	0.38 (<0.001)	1				
10	−0.54 (<0.001)	−0.07 (0.388)	−0.05 (0.564)	−0.49 (<0.001)	−0.42 (<0.001)	−0.59 (<0.001)	−0.53 (<0.001)	−0.37 (<0.001)	−0.31 (<0.001)	1			
11	0.22 (0.004)	0.24 (0.002)	0.07 (0.356)	−0.06 (0.439)	0.14 (0.069)	0.13 (0.089)	−0.02 (0.849)	0.19 (0.013)	0.11 (0.179)	−0.05 (0.524)	1		
12	0.14 (0.083)	0.02 (0.780)	0.12 (0.122)	0.02 (0.826)	0.14 (0.069)	0.09 (0.248)	0.01 (0.951)	0.13 (0.105)	0.05 (0.493)	−0.02 (0.827)	−0.13 (0.109)	1	
13	0.13 (0.108)	0.18 (0.022)	0.01 (0.935)	0.16 (0.039)	0.07 (0.359)	0.04 (0.624)	0.19 (0.014)	0.11 (0.179)	−0.01 (0.913)	−0.17 (0.025)	−0.12 (0.112)	0.18 (0.018)	1
14	0.04 (0.593)	−0.07 (0.406)	−0.02 (0.810)	0.02 (0.791)	0.00 (0.985)	0.11 (0.158)	0.07 (0.400)	0.09 (0.280)	−0.20 (0.010)	0.05 (0.514)	0.12 (0.112)	−0.02 (0.809)	−0.01 (0.937)

Notes. 1. nursing delirium screening scale (Nu-DESC), 2. age (year), 3. days of ICU stay, 4. duration of ventilation use, 5. number of restraints, 6. number of catheters, 7. number of sedative drugs, 8. simplified acute physiology score (SAPS) Ⅲ score, 9. morse fall scale (MFS) score, 10. Glasgow coma scale (GCS) score, 11. pain score, 12. blood urea nitrogen (BUN), 13. c-reactive protein (CRP), 14. white blood cell.

**Table 6 ijerph-20-05889-t006:** Factors affecting the delirium (*n* = 165).

Characteristic	B	S.E.	β	t	*p*	Change in R^2^
(Constant)	2.92	0.46		6.32		
Number of restraints	0.33	0.04	0.46	7.55		0.399
GCS score	−0.19	0.03	−0.34	−5.64		0.092
Days of ICU stay	0.02	0.01	0.14	2.62	0.010	0.022
BUN	0.01	0.00	0.13	2.39	0.018	0.017
F(*p)*	5.70(0.018)					
R^2^ (Adjusted R^2^)	72.8% (53.0%)					
Tolerance	0.80∼0.98					
VIF	1.02∼1.24					
Durbin-Watson	1.95					

Notes. B: unstandardized coefficient, S.E: standard error, β: Standardized Coefficient, GCS: Glasgow coma scale, ICU: Intensive care unit, BUN: blood urea nitrogen, VIF: Variance Inflation Factor.

## Data Availability

Not available.

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
