# Peer review of "Factors Affecting Delirium in ICU Patients"

_ijerph, 2023, doi:10.3390/ijerph20105889_

Round 1

Reviewer 1 Report

There are many studies related to ICU delirium. Factor analysis of delirium has already been done over several years. This study is no longer interesting, and the results of the study do not contribute.

Author Response

Authors’ Response: We appreciate the time and effort that you and the reviewers have put in providing valuable feedback and insightful comments, which improved our manuscript. We have carefully considered each comment and updated the manuscript, as required.

We have marked the revisions made to the manuscript red font.

Point 1: Extensive editing of English language and style required.

Response 1: We appreciate your thoughtful comment. Since we are foreigners in English, we have difficulty conveying meaning in English. To solve this problem, a professional editing company (Editage.co.kr) was requested for editing, and an editing report was attached.

Point 2: There are many studies related to ICU delirium. Factor analysis of delirium has already been done over several years. This study is no longer interesting, and the results of the study do not contribute.

Response 2: Thank you for your valuable feedback. In this study, adult patients who had been admitted to the intensive care unit for 24 hours were classified as delirium patients using two delirium screening tools (CAM-ICU and Nu-DESC). In previous studies, CAM-ICU or Nu-DESC was used only as a screening tool for delirium. However, since Nu-DESC can measure the degree of delirium with a score of 0 to 5, it can be treated as a continuous variable. Therefore, unlike in other studies, correlation and multiple regression analysis between delirium influencing factors and Nu-DESC scores are possible. In this study, the regression analysis results showed that 'days of ICU stay', 'GCS score', and 'BUN level' were the factors influencing delirium, as in previous studies.

However, in this study, there is a difference in analyzing the factors affecting delirium according to the degree of delirium using the Nu-DESC score, not the presence or absence of delirium.

Also, in previous studies, various characteristics such as general, clinical, hemodynamic, and environmental characteristics were separated and analyzed for their relationship with delirium respectively, but in this study, general, clinical, hemodynamic, and environmental characteristics of the ICU were analyzed as influencing factors, it was treated as independent variables at once and analyzed comprehensively.

We believe that for the above reasons, we can fully contribute to the care of delirium patients.

Reviewer 2 Report

Thank you very much for great opportunity for reviewing this manuscript. Overall well written and I would like to ask some poitns.

Major point

1.Conclusion and reccomendation is redundant. Please make clearer messages from the results of this research.

Minor points

1. What is "Levin tube"? Is it naso-gastric tube?

2. In table4, cut off value of "Days of ICU" is 12< and ≧13, which means 12 is not included in both group. Please clarify at this point. 

Author Response

Authors’ Response: We appreciate the time and effort that you and the reviewers have put in providing valuable feedback and insightful comments, which considerably improved our manuscript. We have carefully considered each comment and made changes to the manuscript, as required. We have marked in red font the revisions we made in the manuscript.

Point 1: Conclusion and reccomendation is redundant. Please make clearer messages from the results of this research.

Response 1: We would like to convey our deepest gratitude for your review on our study. We revised Conclusion and recommendations as follows.

  1. Conclusion and Recommendations

This study investigated the incidence and predictors of delirium among patients in the ICU using the Korean version of the Nu-DESC. The results showed that the risk for delirium was higher with a greater number of physical restraints, decreasing GCS score, and increasing ICU LOS and BUN.

Due to the retrospective nature of this study, some factors associated with delirium may not have been examined because they were not available on medical records, such as environmental factors (e.g., lighting, noise, and sleep). Consequently, prospective studies that include environmental factors are needed.

Point 2: What is "Levin tube"? Is it naso-gastric tube?

Response 2: That’s right. We changed it to nasogastric tube.

Point 3: In table 4, cut off value of "Days of ICU" is 12< and ≧13, which means 12 is not included in both group. Please clarify at this point.

Response 3: Thanks for pointing out this valuable point. We modified '≧13' to '≧12'.

Reviewer 3 Report

Thanks for inviting me to review this study exploring prevalence and correlates/predictors of delirium. The paper is a generally well written account of a study that appears scientifically sound.  

I note that while ethical review is reported, there is no reference to consent of patients to participation - in my view this should be addressed with reference to the complexities of recruiting patients who are delirious.  

I also note what seem to me to be particularly high rates of restraint - if patients were so unwell that they needed restraint to prevent removal of tubes, how was their participation secured? 

Author Response

Authors’ Response: We appreciate the time and effort that you and the reviewers have put in providing valuable feedback and insightful comments, which improved our manuscript. We have carefully considered each comment and updated the manuscript, as required.

We have marked the revisions made to the manuscript red font.

Point 1: I note that while ethical review is reported, there is no reference to consent of patients to participation - in my view this should be addressed with reference to the complexities of recruiting patients who are delirious.

Response 1: We appreciate your critical comments. This study was a retrospective study and was approved by the IRB as “subjects were exempted from informed consent”.

Point 2: I also note what seem to me to be particularly high rates of restraint - if patients were so unwell that they needed restraint to prevent removal of tubes, how was their participation secured?

Response 2: Thank you for your valuable feedback. This study was a retrospective study. We collected data from electric medical records.
